# A General Construction Method of Virtual Simulation Experiment Platform Based on Bibliometrics and Analytic Hierarchy Process

**Keyun Zhu** [1], **Juan Cao** [1], **Guowei Chen** [1,*], **Qiang He** [1,2] **and Pengzhou Zhang** [1]

1 State Key Laboratory of Media Convergence and Communication, Communication University of China, Beijing 100024, China
2 State Key Laboratory of Media Convergence Production Technology and Systems, Beijing 100803, China
* Correspondence: cuc_chenguowei@cuc.edu.cn

**Abstract:** Virtual simulation can solve the challenges of high cost, long cycle time, and inaccessibility in traditional experimental teaching, which is far-reaching for talent training. This study combines bibliometric visualization theory with AHP (Analytic Hierarchy Process). It establishes a hierarchical evaluation model of a virtual simulation experimental teaching platform based on 842 questionnaires and 4787 articles, including 68,306 citation records, and deconstructing the complex evaluation problem into several multidimensional factors by attributes and relationships. Based on this, a virtual simulation experimental teaching platform construction scheme for IP protocol analysis based on a network covert communication perspective is outputted, which is compatible with the research results. The experimental platform takes a task-driven teaching method as the core, mainly including four modules of context creation, task determination, independent learning, and effect evaluation. The experience of building this platform can be extended to other disciplines, leading the teaching reform exploration of practice-based, innovation-focused, and engineering-critical, helping to implement the flipped classroom, and promoting the development of education modernization.

**Keywords:** virtual simulation; experimental teaching; analytic hierarchy process; teaching evaluation; bibliometric visualization

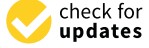



## 1. Introduction

Currently, global higher education is undergoing a milestone change, with the development trend of popularization, internationalization, informatization, and lifelong learning. Experimentation plays a crucial role in higher engineering education, and the virtual simulation experiment platform provides a new perspective for institutional reform and scientific exploration. Establishing a virtual teaching scenario similar to the natural experimental environment can not only effectively solve the technical difficulties in experimental teachings, such as high cost, long lead time, and inaccessibility, but also free students from the limitations of practice opportunities due to lack of equipment, geographical environment, and safety conditions, but directly provide a dynamic and interactive infinite space for students. Especially during the COVID-19 epidemic, when educational institutions around the world were forced to transform online learning methods into the new norm for learning [1], virtual labs were significant and far-reaching.

Since the evaluation factors of virtual simulation experiment platforms are complex, there is no ideal evaluation method yet. This study combines bibliometric theory and AHP to propose an evaluation model to help platform developers use limited resources rationally and maximize benefits and impact.

This study aims to construct an evaluation model based on virtual simulation experimental teaching, which involves knowledge from various disciplines such as computer science, education, management, and statistics. Firstly, a co-occurrence analysis of

4787 pieces of literature, including 68,306 citation records, was conducted using CiteSpace software to visualize the historical evolution of virtual laboratory teaching and its research frontier hotspots. The key factors influencing virtual simulation experimental teaching are derived by observing the node information and the information visualization theory with knowledge measurement. A preliminary structural model is obtained based on their attributes and relationships. Secondly, based on AHP, 842 questionnaires were designed and collected. The evaluation model of the hierarchical virtual simulation experimental teaching platform was established through the steps of data matrix processing, consistency test, and index weight classification calculation. Finally, the evaluation model is applied to a specific field. A construction plan of a virtual simulation experimental teaching platform for IP protocol analysis based on the network covert communication perspective is proposed. The framework structure of this study is shown in Figure 1.

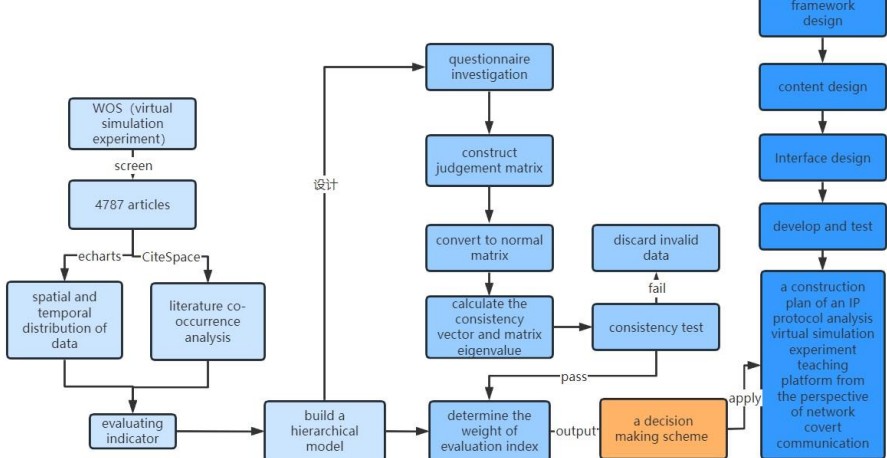

**Figure 1.** Structure of the study.

## 2. Background

### 2.1. Analytic Hierarchy Process

AHP can deconstruct a complex multi-objective decision problem into several levels with multiple indicators to achieve decision optimization.

The original research team of AHP, including Saaty, Vargas, Wind, and others, took eight years from the conception [2] to the precise description [3]. After that, the axiomatic basis [4] was continuously refined to bring it to the academic community. At the same time, it began exploring its application areas, pushing the method toward practical applications, such as marketing [5]. External evaluations of the method have been both positive and negative. For example, Zahedi tested the utility basis of AHP [6] and applied it to a broader field [7]. Dyer argued that AHP was flawed as an alternative program ranking procedure because the procedure produced arbitrary rankings [8].

In 1990, the research team conducted a systematic review of AHP [9,10], opening a new research phase. This phase has several distinctive features.

I.   The increasingly significant cross-fertilization of AHP with other research fields, such as geographic science [11], urban planning [12], social management [13], and education [14].
II.  Increasingly diverse types of problems are used to solve, including selection recommendation [15], evaluation [16], benefit-cost analysis [17], and planning allocation [18].
III. More improved methods are proposed and widely used, such as FAHP [19], which combines fuzzy evaluation with hierarchical analysis, and GAHP [20], which combines gray forecasting with hierarchical analysis.

### 2.2. Virtual Simulation Experiment Platform

The virtual laboratory was first introduced in 1989 by Professor William Wulf of the University of Virginia [21], who described a virtual laboratory environment with networked computers and called it a research center without walls. The earliest web lab was initiated and funded with the participation of the Massachusetts Institute of Technology, allowing students to learn about advanced devices via the Internet [22]. In 1999, UNESCO officially defined the virtual laboratory [23], which gained public recognition and created a research boom.

Initially, virtual laboratories were mainly developed for engineering and later emerged in sociology, art, and other disciplines. The presentation mode of the virtual laboratory is also changing progressively, from web pages with rich pictures and texts to interactive 2D animation and then to a roaming 3D simulation environment. The user experience is becoming better and better. Especially in recent years, with the development of 5G and metaverse, virtual laboratories have been integrated with Augmented Reality, Virtual Reality, and Mixed Reality. The sense of immersion, simulation, and interactivity improved unprecedentedly. Table 1 lists four virtual laboratory projects in different disciplines and different types of presentations.

**Table 1.** Representative virtual laboratory projects.

| Project | Organization | Field | Introduce | Website |
|---|---|---|---|---|
| LiLa | University of Stuttgart, Germany | currency | Launched by eight universities and three companies and funded by the European Commission, LiLa has built a portal that provides access to virtual laboratories [24]. | https://www.lila-project.org/ (accessed on 4 January 2023). |
| Go-Lab | University of Twente, The Netherlands | currency | Go-Lab is a global online science laboratory for inquiry-based learning in schools, funded by the European Commission and uniting 19 organizations from 12 countries. Universities can use these online labs, schools, teachers, and students to extend regular learning activities through science experiments [25]. | https://www.golabz.eu/ (accessed on 4 January 2023). |
| TEALsim | Massachusetts Institute of Technology, USA | Physics | TEALsim is an open-source environment that aims to improve the conceptual and analytical understanding of the nature and dynamics of electromagnetic phenomena by enabling students to see the unpredictable magnetic field lines that are not available in the real world [26]. | http://web.mit.edu/viz/soft/visualizations/tealsim/index.html (accessed on 4 January 2023). |
| Second Life | Linden Labs, USA | Language learning | Second Life allows people to create avatars and interact with other users and user-created content in a multiplayer online virtual world. It is currently the most mature and popular multi-user educational virtual world platform. Language learning is one of the most prevalent types of education [27]. | https://secondlife.com/ (accessed on 4 January 2023). |

As can be seen from the above background, the current virtual laboratories have some shortcomings.

I. Due to the technical complexity of different experiments, most virtual labs are discipline-oriented and not generic, and thus also lack a common evaluation standard.
II. Virtual labs mostly restore experimental steps but lack a virtual environment, i.e., a real atmosphere and no communication path between users.

Initially, the application scenarios of virtual labs were focused on engineering, robotics, and physics teaching. Gradually, virtual labs began to support collaborative learning, user management, inter-user communication, student assignment delivery, grade management, etc. Since 2010, there has been an increasing number of articles discussing the effectiveness of virtual labs in education. At the same time, keywords such as 3D modeling and serious games are gradually increasing in the articles. Scholars are beginning to focus on using 3D immersive environments to enhance learning [28]. In order to support the smooth

operation of 3D environments, web services and cloud computing technologies have also emerged as key technologies.

The explosion and rapid spread of COVID-19 made it difficult for higher education to create a hands-on component of the curriculum. BCcampus has created a directory of virtual laboratory science resources to support distance science education under the Creative Commons Attribution 4.0 International License.

There has been debate in the literature as to whether virtual experiments can replace or enhance students' performance in real labs. For example, it has been argued that manual laboratories are essential for acquiring tactile skills and instrumental awareness but that this is difficult to obtain through virtual laboratory platforms [29]. Scholars have recently researched various approaches, such as experimental design, achievement testing, participant observation, and semi-structured interviews, with a growing number of proponents [30]. It is still worthwhile to explore in depth how to design effective learning environments that are more suitable for the characteristics of digital-age learners.

### 2.3. Bibliometrics

The early discussion of file metrics began in the 1950s [31]. In recent years, the field has developed rapidly with the advent of scientific databases such as Scopus and Web of Science, and bibliometric software such as Citespace, Gephi, Leximancer, and VOSviewer.

The contents of the bibliometric analysis include citation analysis, co-citation analysis, coupling analysis, and so on. The details are shown in Table 2.

**Table 2.** Main contents of bibliometric analysis.

| Analysis Content | Proposed Time | Founder | Introduce |
|---|---|---|---|
| Citation Analysis | 1966 | Eugene Garfield | The citation relationships between publications were analyzed by identifying the most influential publications in the field of study. |
| Co-citation of documents | 1973 | Small | When one or more later papers simultaneously cite two (or more) papers, they constitute a literature co-citation relationship. |
| Author co-cited | 1981 | White, Griffith | When one or more later papers simultaneously cite two (or more) authors, they constitute a literature co-citation relationship. |
| Document coupling | 1963 | M.M.Kessler | If literature A and B cite the same reference, they constitute a coupling relationship. |

Bibliometrics can be applied to various fields to study the current state of the discipline and its future directions. It can be either a large area, such as machine learning [32], or a small direction, for example, studying machine learning and soft computing applications in the textile and apparel supply chain [33].

## 3. Establishment of a Hierarchical Model Based on CiteSpace

### 3.1. Methods and Data

3.1.1. Technology Roadmap

The purpose of this study is to obtain a generic approach to the construction of a virtual simulation experiment platform. Therefore a complete extraction of the influencing factors is required. Many scholars have studied it based on different perspectives so that the analysis can be carried out and factors can be obtained based on the past literature.

Bibliometric visualization enables the study of the direction of domain evolution and the transformation of the knowledge domain from an unknown competitive paradigm to a detectable one, a fast-developing means of literature data processing in recent years. Chaomei Chen, one of the earliest pioneers in the field of information visualization, has been developing applications that can visualize knowledge information since the 1990s, creatively combining information visualization techniques with scientometrics. In 2004, Chaomei Chen developed CiteSpace, a software designed to conduct information visualization studies in scientific fields. By analyzing the co-occurrence of information such as titles, authors, keywords, and citations of literature [34], the knowledge domain can be visualized and studied. CiteSpace gradually became a new tool commonly adopted in the field of scientometrics [35].

### 3.1.2. Data Retrieval

In order to obtain comprehensive results, this study was limited to the last 20 years, i.e., 2002 to 2022. A search of the Web of Science core collection with the subject "virtual simulation experiment" or "virtual lab" yielded 5772 records. The Web of Science core collection was searched with 5772 records containing rich citation information. In order to avoid omissions due to inaccurate keywords, this paper was supplemented with 94 data obtained from the virtual laboratory-related review literature [36–38].

### 3.1.3. Data Filtering and Aggregation

The above literature was further reviewed, and articles that did not meet the criteria were excluded. Specific requirements were that the literature ranged from 2002 to 2022, was written in English, and the content was related to virtual laboratories, including but not limited to construction methods, application cases, and effectiveness evaluation.

After screening and aggregation, 4787 articles were obtained, including 68,306 citation records. The complete records containing cited references were exported in plain text format. The original research database was established based on this.

### 3.1.4. Data Standardization

Standardization of the literature data is necessary, such as standardizing the writing of the same author's name, different names of the same journal, etc. This step can prepare for the following analysis.

### 3.2. Spatial and Temporal Distribution of Data

As shown in Figure 2, the exploratory research on virtual imitation experimental teaching has been relatively short. In general, the volume of papers remained at a low level from 2002 to 2014, but after that, rapid growth was presented. From 2017 to the present, the volume of publications has remained stable. In terms of development background, from 1999 to 2013, online teaching was still at a low level. With the continuous development of the Internet, virtual simulation experimental teaching has gradually received the attention of the government and enterprises, and a large number of relevant academic achievements have been born.

The spatial distribution of the literature is helpful for scholars to quickly identify the distribution of significant research power in the world and provide critical practical references for scientific research communication and cooperation. In this paper, the regional fields of the literature are extracted. The top 13 regions of the paper output are shown in Figure 2. From the overall statistical distribution, the research output of virtual simulation experimental teaching is uneven globally. The high-output regions tend to be more economically developed or pay more attention to education and Internet development.

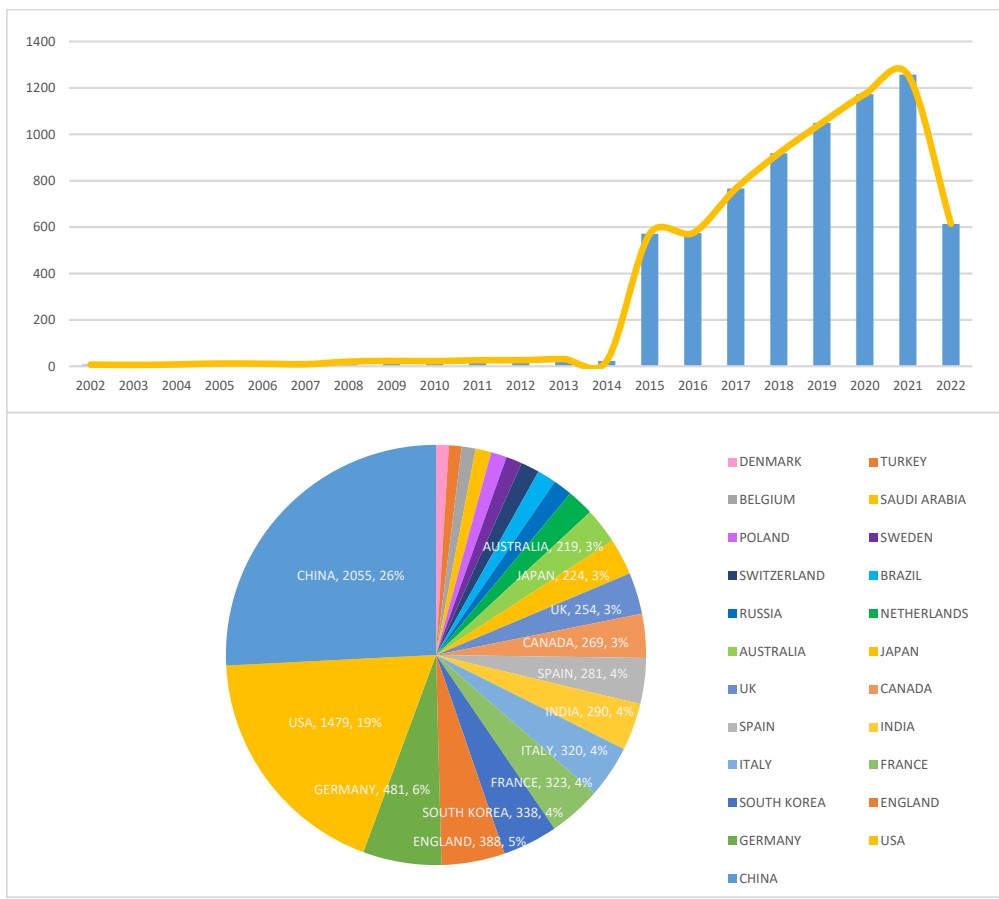

**Figure 2.** Spatial and temporal distribution of data.

### 3.3. Literature Co-Occurrence Analysis

The co-occurrence analysis method proposed by M. Callon et al. refers to the correlation between one thing and another or more things in terms of the way they are presented externally [39]. The co-occurrence discussed in this thesis is the co-occurrence between knowledge units of scientific and technical texts, such as the co-occurrence between different keywords, the co-occurrence between different authors in the same paper, and the co-occurrence between different references.

The results of the statistics on the topics with frequencies greater than or equal to 100 occurrences are shown in Table 3. The theme simulation appeared 795 times as a basic vocabulary. The classification of other terms based on lexicality and lexical meaning yielded hot research areas in virtual simulation experimental teaching, such as dynamics, behavior, optimization, and numerical simulation, as well as popular research methods, such as modeling, virtual reality, and algorithms.

Clustering is the process of grouping the original data into different clusters. Objects in the same cluster have a high degree of similarity, while objects differ significantly between different clusters. After clustering the results of the above-mentioned thematic co-occurrence and generating visual plots, the co-citation network clusters are adjusted. After obtaining a more satisfactory mapping, the clusters were named using the LLR algorithm and finally shown in Figure 3 (left). In order to explore the research contents and hotspots in different periods, this paper adopts the timeline presentation to get the visualization map, as shown in Figure 3 (right).

**Table 3.** Subject co-occurrence analysis statistics.

| Frequency | First Occurrence Time | Term |
|---|---|---|
| 795 | 1999 | simulation |
| 368 | 1999 | model |
| 309 | 2002 | design |
| 280 | 1999 | system |
| 238 | 1999 | virtual reality |
| 181 | 2003 | performance |
| 148 | 1999 | algorithm |
| 146 | 2000 | dynamics |
| 142 | 2003 | behavior |
| 128 | 2006 | optimization |
| 116 | 2012 | cloud computing |
| 101 | 2000 | numerical simulation |

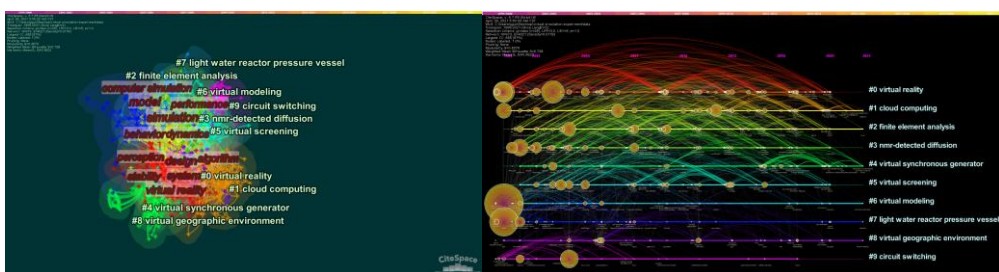

**Figure 3.** Results of the thematic co-occurrence clustering analysis (**left**), the timeline presentation (**right**).

According to the graph analysis, the related research of virtual laboratories revolves around virtual reality, cloud computing, computer simulation, virtual modeling, and so on. "Computer simulation" indicates that researchers often attach importance to the degree of simulation of virtual laboratories and devote themselves to improving simulation accuracy to solve practical problems. "Cloud Computing" indicates that researchers pay attention to the fluency of the experimental platform and solve the problem of running the platform through distributed computing. "Virtual reality" shows that researchers pay attention to the interactivity and immersion of the forum and bring a better user experience through richer forms and more exquisite modeling.

The citation co-occurrence analysis can obtain the high-impact literature in the field, which has similar research topics and is a liaison to different topics, etc. Based on the above literature, citation contribution analysis was performed, and the results were obtained, as shown in Figure 4. More than ten essential documents, including CloudSim: a toolkit for modeling and simulation of cloud computing environments and evaluation of resource provisioning algorithms from RN Calheiros in 2011, were obtained, and read carefully, to further abstract the key elements.

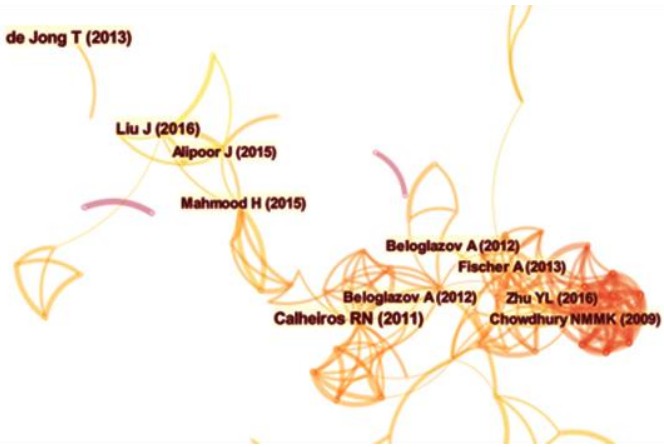

**Figure 4.** Citation co-occurrence results.

### 3.4. Hierarchy Construction

AHP for decision problems must first deconstruct the complex problem into elements formed by attributes and spatial relationships at different levels, where the upper-level elements dominate the lower-level elements. These layers are generally subdivided into objective, criterion, and solution layers. Based on the above visualization results, the evaluation elements of the virtual simulation experiment platform proposed in this thesis are shown in Figure 5.

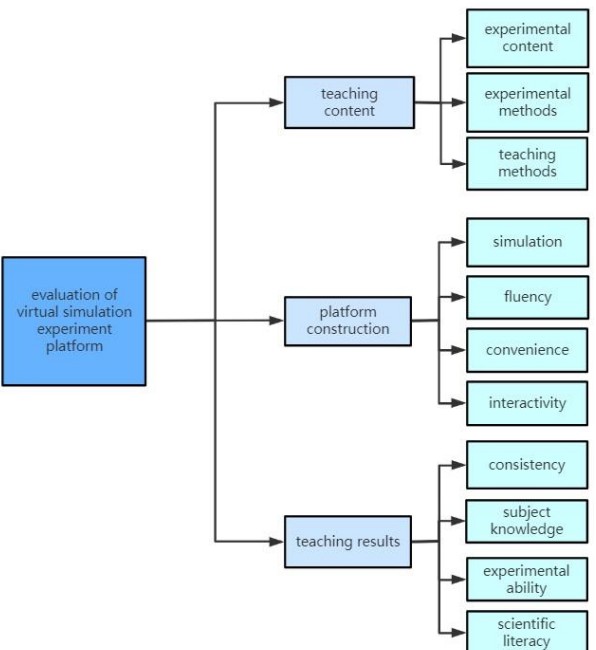

**Figure 5.** Evaluation elements of the virtual simulation experiment platform.

The target layer of the hierarchical model is the evaluation of the virtual simulation experiment platform, and the guideline layer is divided into three parts: teaching content, platform construction, and teaching results. The program layer based on teaching content includes experimental content, experimental methods, and teaching methods. The program layer based on platform construction includes simulation, fluency, convenience, and interactivity. The program layer based on teaching results includes consistency, subject knowledge, experimental ability, and scientific literacy. Each program layer is further set up with specific observation points, as shown in Table 4.

**Table 4.** Observation points of the evaluation model.

| Target Layer | Guideline Layer | Program Layer | Observation Points |
|---|---|---|---|
| evaluation of virtual simulation experiment platform A | teaching content A1 | experimental content A11 | meet the requirements of talent training and syllabus, take into account the depth and breadth, adapt to the needs of students at different levels, integrate the latest achievements of the discipline industry development |
| | | experimental methods A12 | the experimental process is complete and standardized, the experimental process is multi-way, the experimental results are unique |
| | | teaching methods A13 | autonomy, innovation |
| | platform construction A2 | simulation A21 | simulation degree; solving practical problems |
| | | fluency A22 | platform running lag situation, clear guidance, easy to operate the process, repeatability |
| | | convenience A23 | experimental platform software and hardware requirements, redundancy time, network requirements |
| | | interactivity A24 | interest, flexibility, expansibility |
| | teaching results A3 | consistency A31 | relevance of experimental platform evaluation results to reality |
| | | subject knowledge A32 | knowledge mastery, desire for inquiry |
| | | experimental ability A33 | problem thinking and analysis skills, hands-on skills, experimental investigation skills |
| | | scientific literacy A34 | creative consciousness, rational thinking, independent thinking, moral level, social responsibility |

## 4. Determination of Evaluation Index Weights

### 4.1. Judgment Matrix

Sometimes we need to compare the influence of n factors X = {x1, . . . , xn} on one factor Z. In order to provide more credible data, we can consider pairwise comparison matrices. That is, two factors xi and xj are selected each time, and the ratio of the influence of xi and xj on Z is recorded as *aij*. The result is replaced using a matrix $A = (aij)^{n*n}$, and *A* is called the judgment matrix between Z and X. It is easy to see that if the ratio of xi to xj on Z is *aij*, the ratio of xj to xi on Z should be $aji = \frac{1}{a_{ij}}$. Regarding determining the value of *aij*, the numbers 1 to 9 and their reciprocals are quoted as scales. The meanings of 1 to 9 are listed in Table 5, respectively.

In this study, data were collected through questionnaires. The questionnaires were distributed to users through online and offline methods. The planned research objects are 50% of students, 10% of teachers, and 40% of practitioners in other industries. Offline visits to colleges and universities and their surrounding business districts, online through the "Questionnaire Star" software, fully mobilize the enthusiasm of the majority of netizens and control the quality of questionnaires through manual intervention. The questionnaire is divided into three major parts. The first part is the basic information of the questionnaire filler, with two questions; the second is the first level of indicator comparison, with one question; the third is the second level of indicator comparison, with three questions.

A small sample survey step was added to the development stage of the questionnaire to test whether the respondents could answer effectively, and the survey respondents mainly included college students and teachers. The data recovered from the questionnaire did not reflect significant problems, and the answers met the expectations of this paper, so the formal questionnaire survey was started, and 842 questionnaires were finally obtained.

Among them, 37.06% of the respondents have used the virtual experiment platform, while the rest of the users have not used it.

**Table 5.** Criteria for assignment of elements in a judgment matrix.

| *aij* | Definition | *aij* | Definition |
|-------|------------|-------|------------|
| 1 | *ai* and *aj* are equally important | 2 | Somewhere between equal and slightly more important |
| 3 | *ai* is slightly more important than *aj* | 4 | Somewhere between slightly more important and significantly more important |
| 5 | *ai* is significantly more important than *aj* | 6 | Somewhere between significantly more important and strongly more important |
| 7 | *ai* is strongly more important than *aj* | 8 | Somewhere between strongly more important and extremely more important |
| 9 | *ai* is extremely more important than *aj* | | |

The questionnaire data are preprocessed, and the results of pairwise comparisons of *n* elements of the questionnaire results are placed in the upper triangular part of the comparison matrix *A*, as shown in Equation (1). The value of the lower triangular part is the reciprocal of the value of the relative position of the upper triangle. That is, $aji = \frac{1}{a_{ij}}$.

$$A = \begin{bmatrix} a_{ij} \end{bmatrix} = \begin{bmatrix} 1 & a_{12} & \cdots & a_{1n} \\ \frac{1}{a_{12}} & 1 & \cdots & a_{2n} \\ \vdots & \vdots & & \vdots \\ \frac{1}{a_{1n}} & \frac{1}{a_{2n}} & \cdots & 1 \end{bmatrix} \tag{1}$$

As an example, the comparison matrix was constructed for the first three data items, and the results are shown in Table 6 below.

**Table 6.** Questionnaire data matrix (first 3 rows).

| Number | Teaching Content | Platform Construction | Teaching Results |
|--------|------------------|----------------------|------------------|
|  | 1.00 | 3.00 | 1.00 |
| 1 | 0.33 | 1.00 | 0.33 |
|  | 1.00 | 3.00 | 1.00 |
|  | 1.00 | 1.00 | 1.00 |
| 2 | 1.00 | 1.00 | 1.00 |
|  | 1.00 | 1.00 | 1.00 |
|  | 1.00 | 7.00 | 9.00 |
| 3 | 0.14 | 1.00 | 9.00 |
|  | 0.11 | 0.11 | 1.00 |

*4.2. Data Analysis*

In calculating vector values by hierarchical analysis, Saaty proposed a row vector mean normalization method to calculate them. Since most of the matrices in the calculation of this optimization method are non-consistent, better accuracy can be obtained by applying this optimization method.

Equation (2) transforms the original matrix into a regular matrix *W*.

$$W_i{}' = \frac{1}{n} \sum_{j=1}^{n} \frac{a_{ij}}{\sum_{i=1}^{n} a_{ij}} \qquad i, j = 1, 2, \ldots\ldots, n \tag{2}$$

From Equation (3) and the regular matrix *W*, the consistency vector *v* can be calculated.

$$v_i = \frac{\sum_{j=1}^{n} w_j a_{ij}}{w_i} \qquad i,j = 1,2,\ldots\ldots,n \tag{3}$$

After obtaining the consistency vector *v*, the matrix characteristic root $\lambda$ is obtained from Equation (4).

$$\lambda_i = \frac{\sum_{i=1}^{n} v_i}{n} \qquad i,j = 1,2,\ldots\ldots,n \tag{4}$$

The matrix of the criterion layer was processed, and the first three data results are shown in Table 7.

**Table 7.** Results of questionnaire data matrix processing (first 3 rows).

| | A1 | A2 | A3 | Normalized Matrix | | | $W_i'$ | $v_i$ | $\lambda_i$ |
|---|---|---|---|---|---|---|---|---|---|
| | 1.00 | 3.00 | 1.00 | 0.4286 | 0.4286 | 0.4286 | 0.4286 | 1.2857 | 3.0000 |
| 1 | 0.33 | 1.00 | 0.33 | 0.1429 | 0.1429 | 0.1429 | 0.1429 | 0.4286 | 3.0000 |
| | 1.00 | 3.00 | 1.00 | 0.4286 | 0.4286 | 0.4286 | 0.4286 | 1.2857 | 3.0000 |
| | 1.00 | 1.00 | 1.00 | 0.3333 | 0.3333 | 0.3333 | 0.3333 | 1.0000 | 3.0000 |
| 2 | 1.00 | 1.00 | 1.00 | 0.3333 | 0.3333 | 0.3333 | 0.3333 | 1.0000 | 3.0000 |
| | 1.00 | 1.00 | 1.00 | 0.3333 | 0.3333 | 0.3333 | 0.3333 | 1.0000 | 3.0000 |
| | 1.00 | 7.00 | 9.00 | 0.7975 | 0.8630 | 0.4737 | 0.7114 | 2.8350 | 3.9851 |
| 3 | 0.14 | 1.00 | 9.00 | 0.1139 | 0.1233 | 0.4737 | 0.2370 | 0.8034 | 3.3904 |
| | 0.11 | 0.11 | 1.00 | 0.0886 | 0.0137 | 0.0526 | 0.0516 | 0.1570 | 3.0403 |

*C.I.* is defined as the consistency index, as shown in Equation (5). *C.I.* = 0 indicates perfect consistency, and Saaty suggests that all cases with *R.I.* < 0.1 are considered to have a better consistency.

$$C.I. = \frac{\lambda - n}{n - 1} \tag{5}$$

The research of Dak Ridge National Laboratory and Wharton School showed that the matrix of assessment scales 1 to 9, at different strata numbers, produces different consistency indicators (*C.I.*) and random indicators (*R.I.*). The ratio of *C.I.* values to *R.I.* values, called the consistency ratio (*C.R.*), is shown in Equation (6).

$$C.R. = \frac{C.I.}{R.I.} \tag{6}$$

Therefore, when the *C.R.* value is less than 0.1, the degree of consistency of the matrix is high. Its random indicator values are shown in Table 8 below.

**Table 8.** Random indicators.

| Step | 1 | 2 | 3 | 4 | 5 | 6 | 7 | 8 |
|---|---|---|---|---|---|---|---|---|
| *R.I.* | 0.00 | 0.00 | 0.58 | 0.90 | 1.12 | 1.24 | 1.32 | 1.41 |
| Step | 9 | 10 | 11 | 12 | 13 | 14 | 15 | |
| *R.I.* | 1.45 | 1.49 | 1.51 | 1.48 | 1.56 | 1.57 | 1.58 | |

Based on the above formula, the matrix of the criterion layer is calculated for consistency, and the results of the first three data are shown in Table 9.

**Table 9.** Data consistency (first 3 rows).

| | A1 | A2 | A3 | $W_i'$ | $v_i$ | $\lambda_i$ | C.I. | C.R. | Value |
|---|---|---|---|---|---|---|---|---|---|
| | 1.00 | 3.00 | 1.00 | 0.4286 | 1.2857 | 3.0000 | | | |
| 1 | 0.33 | 1.00 | 0.33 | 0.1429 | 0.4286 | 3.0000 | 0.0000 | 0.0000 | valid |
| | 1.00 | 3.00 | 1.00 | 0.4286 | 1.2857 | 3.0000 | | | |
| | 1.00 | 1.00 | 1.00 | 0.3333 | 1.0000 | 3.0000 | | | |
| 2 | 1.00 | 1.00 | 1.00 | 0.3333 | 1.0000 | 3.0000 | 0.0000 | 0.0000 | valid |
| | 1.00 | 1.00 | 1.00 | 0.3333 | 1.0000 | 3.0000 | | | |
| | 1.00 | 7.00 | 9.00 | 0.7114 | 2.8350 | 3.9851 | | | |
| 3 | 0.14 | 1.00 | 9.00 | 0.2370 | 0.8034 | 3.3904 | 0.2360 | 0.4068 | invalid |
| | 0.11 | 0.11 | 1.00 | 0.0516 | 0.1570 | 3.0403 | | | |

The data relating to the criterion level were processed, and the results were obtained as shown in Table 10, where the average contribution of teaching content to the construction of the virtual simulation experiment platform was nearly half. Users who have used the virtual simulation experiment platform pay more attention to the teaching results. Users who have not used the virtual simulation experiment platform pay more attention to the construction of the platform.

**Table 10.** Average contribution of questionnaire data by usage.

| Usage | A1 | A2 | A3 |
|---|---|---|---|
| Yes | 0.4706 | 0.2073 | 0.3221 |
| Sort | 1 | 3 | 2 |
| No | 0.4382 | 0.3063 | 0.2555 |
| Sort | 1 | 2 | 3 |

From Table 11, students pay more attention to the teaching content of the virtual simulation experiment platform and pay similar attention to the construction of the platform and teaching outcomes. Teachers think the teaching content and teaching outcomes of the virtual simulation experiment platform are similar, while platform construction is relatively unimportant. Administrators think teaching content and platform construction are more critical while teaching outcomes are less influential.

**Table 11.** Average contribution of questionnaire data by occupation.

| (2) Occupation | A1 | A2 | A3 |
|---|---|---|---|
| Student | 0.4616 | 0.2317 | 0.3067 |
| Sort | 1 | 3 | 2 |
| Teacher | 0.4796 | 0.115 | 0.4055 |
| Sort | 1 | 3 | 2 |
| Administrator | 0.5362 | 0.2792 | 0.1847 |
| Sort | 1 | 2 | 3 |
| Technician | 0.3333 | 0.3333 | 0.3333 |
| Sort | 1 | 1 | 1 |
| Else | 0.3351 | 0.4446 | 0.2203 |
| Sort | 2 | 1 | 3 |

*4.3. Algorithm Design*

The above uses Excel to calculate indicator weights for the criteria layer. Although this method is feasible, it requires a lot of time and effort and is error-prone. Based on this, this section designs a Python-based algorithm that implements the reading, storing, and automatic calculation of the table data and related data.

The algorithm needs to call CSV, Numpy, and Pandas library files, and the specific functions of the modules are as follows.

CSV is a class that implements reading and writing tabular data in CSV format. It allows the programmer to read data directly from the document generated by Excel while hiding the implementation details of Excel using CSV format to process data.

Numpy, with strong multi-dimensional vector model arithmetic analysis capabilities, provides Python support for multi-dimensional arrays and vector objects. It can support a large number of matrix operations of different dimensions at the same time, in addition to providing an extensive library of mathematical functions.

Pandas, a powerful toolset for analyzing structured data, is based on Numpy. It is used for data mining and analysis and provides data cleaning functions.

In turn, the input data are computed with feature values, feature vectors, RI matrix, and consistency, which can determine the validity of the data and output indicator weights, as shown in Algorithm 1.

---

**Algorithm 1.** Analytic hierarchy process.

---

Input: original matrix
Output: Maximum eigenvalue and corresponding eigenvector, consistency test results

1.  class AHP:
2.  def Initialization():
3.  Record matrix size
4.  Initialize RI for consistency check
5.  def Maximum eigenvalue and Corresponding eigenvector ():
6.  Use numpy.linalg.eig() Function to calculate the eigenvalues and eigenvectors of the matrix
7.  Determine the eigenvector corresponding to the maximum eigenvalue
8.  Add maximum eigenvalue attribute
9.  Calculate weight vector W
10. return Maximum eigenvalue and corresponding eigenvector
11. def Consistency test ():
12. Calculate CI
13. Classified discussion
14. if rank == 2:
15. No consistency problem
16. else:
17. Calculate CR
18. if CR < 0.10:
19. Pass the consistency test
20. return True
21. else:
22. Failure to pass the conformance test

return False

---

*4.4. Decision Scheme*

The synthetic weights of the index system can be derived by calculating the weights between the indicators at each level of the index system relative to the overall volume and target weights, as shown in Table 12 below.

**Table 12.** Synthetic weight for the evaluation indicators.

| Primary Index | Secondary Index | Synthetic Weight |
|---|---|---|
| teaching content 0.4576 | experimental content 0.5547 | 0.2538 |
| | experimental methods 0.1920 | 0.0879 |
| | teaching methods 0.2533 | 0.1159 |

**Table 12.** *Cont.*

| Primary Index | Secondary Index | Synthetic Weight |
|---|---|---|
| platform construction 0.2469 | simulation 0.1321 | 0.0326 |
| | fluency 0.4402 | 0.1087 |
| | convenience 0.1723 | 0.0425 |
| | interactivity 0.2454 | 0.0606 |
| | consistency 0.1272 | 0.0376 |
| teaching results 0.2955 | subject knowledge 0.4896 | 0.1447 |
| | experimental ability 0.2596 | 0.0767 |
| | scientific literacy 0.1236 | 0.0365 |

## 5. A Virtual Imitation Platform Case Based on Evaluation Models

This study presents a case study of building a domain-specific virtual simulation experiment platform based on the above research results. The experimental platform is designed around task-driven pedagogy, integrating constructivist theory, Kolb's learning circle, and other pedagogical theories. It is divided into four modules: creating a situation, defining a task, independent learning, and evaluating the effect. After entering the system, users will be guided by a video to accept the task and detect the hidden communication content. Subsequently, basic, advanced, and innovative experiments are conducted to complete independent learning and solve the task. Finally, the platform generates an experiment report to evaluate the effectiveness of the user's learning. The framework of the experimental platform is shown in Figure 6.

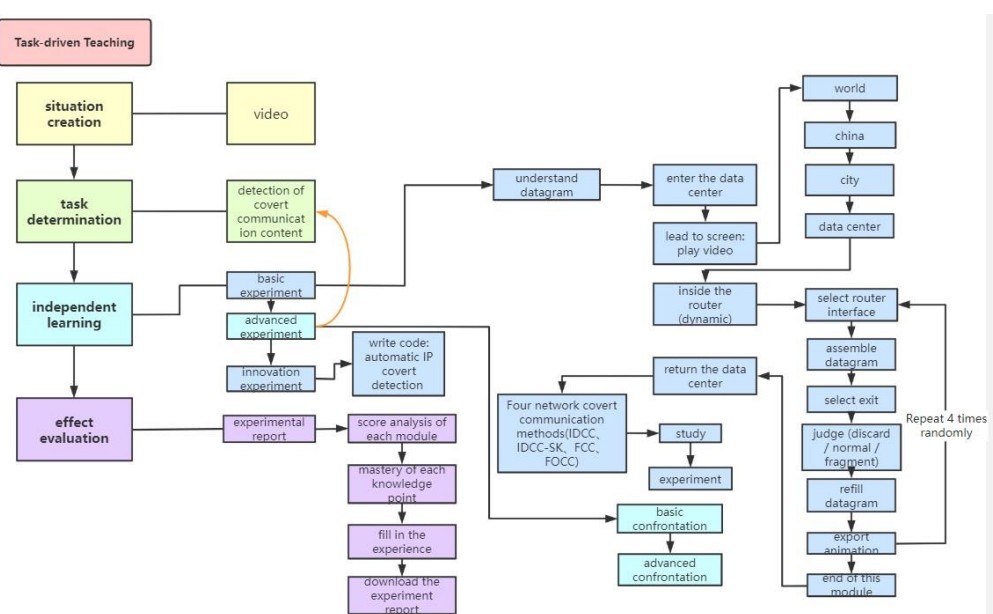

**Figure 6.** Framework of the experimental platform.

Based on the above design, a virtual simulation experimental teaching platform for IP protocol analysis based on network covert communication is constructed, and its operation interface is shown in Figure 7.

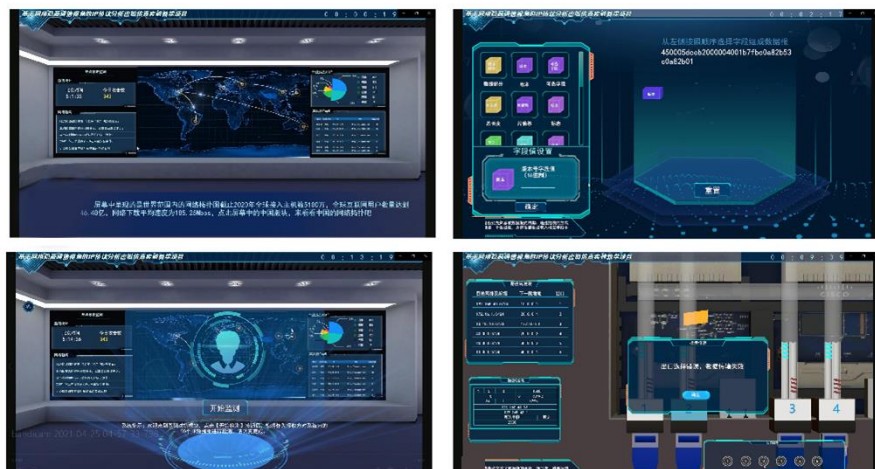

**Figure 7.** Screenshot of experimental platform.

This platform uses Procreate and Photoshop to draw the UI interface, including the startup interface, guiding characters, buttons, etc. It uses 3Dds Max to build 3D models, including cyber police center and data center scenes, experimental props such as routers and fire extinguishers, and abstract elements such as IP datagrams. The dynamic effects of the platform are made using software such as AfterEffects and Premiere, including holographic screens, global network topology, and the movement process of IP datagrams in routers, etc. Unity3d is used to develop the experimental platform, completing the integration of the 3D scene display control module and the network communication module. Specifically, it includes 3D model creation and import, motion trajectory setting, unique effect production, particle system, collision detection, etc.

Users can learn IP protocol analysis through this platform. For example, in the datagram building module, an IP datagram can be built by dragging and dropping. The learner judges the output port of the router according to the header information and the routing table, and the datagram flows along the path to the exit and the port, and datagram information pops up.

## 6. Conclusions

On the one hand, the research results of this thesis can provide strategic support for constructing a virtual simulation experimental teaching platform. On the other hand, the discussion of the evaluation model can provide adequate decision support for the development of education modernization and promote the development of flipped classrooms.

I. Innovatively introduce the bibliometric visualization theory into the study of virtual simulation experimental teaching. Using CiteSpace software, the evolution of virtual imitation experimental teaching and its research frontiers and hotspots are visualized through the literature co-occurrence analysis. By observing the visualization nodes and combining the information visualization theory of knowledge measurement, the key factors affecting the teaching of virtual imitation experiments are obtained.

II. An AHP-based evaluation model of a virtual simulation experimental teaching platform is proposed. The target layer is the evaluation of the virtual simulation experimental platform, and the guideline layer is divided into three parts: teaching content, platform construction, and teaching results. The program layer based on teaching content includes experimental content, experimental method, and teaching method. The program layer based on platform construction includes simulation degree, fluency, convenience, and interactivity. The program layer based on teaching results includes the unification of attainment evaluation, subject knowledge, experimental ability, and scientific literacy.

III. Algorithm design based on Python to complete the intelligent calculation of relevant indexes of hierarchical analysis method, mainly including data access and consistency

test. Based on the program running results, the weights of each evaluation index are obtained, and an evaluation model of a virtual imitation experimental teaching platform is constructed.

IV.  Based on the evaluation model, a virtual simulation experimental teaching platform construction program from the network covert communication perspective of IP protocol analysis is output with the research results. The experience of building this virtual simulation experiment platform can be extended to other disciplines to guide them. It leads the teaching reform exploration of practice-based, innovation-oriented, and engineering-oriented, and promotes the development of education modernization.

**Author Contributions:** Conceptualization, K.Z., J.C., G.C., Q.H. and P.Z.; methodology, K.Z., J.C., G.C., Q.H. and P.Z.; software, K.Z., J.C., G.C., Q.H. and P.Z.; validation, K.Z., J.C., G.C., Q.H. and P.Z.; formal analysis, J.C.; investigation, Q.H.; resources, G.C.; data curation, K.Z., J.C., G.C., Q.H. and P.Z.; writing—original draft preparation, K.Z.; writing—review and editing, Q.H.; visualization, K.Z.; supervision, P.Z.; project administration, J.C.; funding acquisition, J.C., G.C. and P.Z. All authors have read and agreed to the published version of the manuscript.

**Funding:** This work was supported by the National Key Research and Development Program of China (2020AAA0108700), the Fundamental Research Funds for the Central Universities (CUC220C004), and the State Key Laboratory of Media Convergence Production Technology and Systems (SKLM-CPTS2020012).

**Informed Consent Statement:** Informed consent was obtained from all participants involved in the study.

**Data Availability Statement:** Data are available upon request to the corresponding author.

**Conflicts of Interest:** The authors declare no conflict of interest.

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
