# Peer review of "A General Construction Method of Virtual Simulation Experiment Platform Based on Bibliometrics and Analytic Hierarchy Process"

_education, doi:10.3390/educsci13010080_

Round 1

Reviewer 1 Report

This is a complex paper which proposes combining bibliometric visualization theory with the Analytic Hierarchy Process, aiming to construct an evaluation model based on virtual simulation experimental teaching. It includes a large (and complex) body of work, for which we commend the authors.
Various practical observations are highlighted. For instance (as resulting from the data shown in Table 11), the authors importantly observed that students, teachers and administrators pay attention to different aspects of the virtual platforms.

Overall, this is important work and the reviewer only has a few minor remarks, as below.

- Table 1: "representative virtual laboratory projects" - how did the authors choose these 4 examples (and were there not more representative/popular examples of such projects used in the world today?)

- subsections 3.1.3 and 3.1.4 are both named "Data filtering and aggregation"
- The "results of the thematic co-occurrence clustering analysis" need to be moreclearly explained (Figure 3)
- Figure 4 is hardly visible

- line 247: "In this study, data were collected through questionnaires" - It is important to make it clear exactly how/to whom etc. these questionanires were administered 

 - Section 5 shows in detail a framework of the proposed experimental platform (Figure 6). We commend the authors for their complex framework structure.
This section concludes with the following sentence "Based on the above design, a virtual simulation experimental teaching platform for IP protocol analysis based on network covert communication is constructed.". In fact, this "platform for IP protocol analysis" is only mentioned here and the introduction&conclusions. Could the authors, please, clarify?

Similarly, part IV of the conclusions mention "the design of Civic and Political integration points". What do the authors refer to exactly?

One last thing: would it be possible to maybe propose a more suggestive title? ("According to..." sounds slightly unnatural after the column in the title...)

Reviewer 2 Report

There are several typo errors around line 240-245 in formulae.

Also it needs spelling check on it is if instead of "of'

Paper is well done.

There could have a little more mention of Virtual reality and zoom software. How they come in the play for education.

Social sciences may be user and may not need heavy equipment as physical sciences need.

These days all depts need virtual labs in one form or the other other. Computer Engineering may need virtual lab, computer science virual reality can be taught without headset etc, but user may use head set using real app as virtual reality in the context of  head sets vs hololense etc.

Anyway, I liked the paper content.
